# Quantitative Study on Agricultural Premium Rate and Its Distribution in China

Yaoyao Wu, Hanqi Liao, Lei Fang and Guizhen Guo *

National Disaster Reduction Center of China (NDRCC), Ministry of Emergency Management, Beijing 100124, China
* Correspondence: guoguizhen@ndrcc.org.cn

**Abstract:** In recent years, with the deepening of the reform of rural economic systems, the demand for disaster risk governance in land production and management is increasing, and it is urgent for the state to develop agricultural insurance to improve land production recovery capacity and ensure national food security. The study develops a quantitative model to determine the agricultural premium rate for each county in China based on disaster risk level in order to refine agricultural insurance. The results show that: (a) in terms of the disaster situation, most of northeast and central China, part of southwest, north, and northwest China are seriously affected; (b) regarding the integrated natural disaster risk level, there are 129 counties with extremely high disaster risk in China; (c) as for agricultural premium rates based on the integrated natural disaster risk index, some counties in Inner Mongolia, Shanxi, Liaoning, Jilin, Shandong, Anhui, Jiangxi, Zhejiang, Guangdong, Hubei, and Hunan Province had extremely high rates, out of a total of 63 counties. The above results reveal regional differences in disaster risk levels and premium rates between counties, providing a reference for improving the accuracy of agricultural premium rates. This contributes to the creation of security for further improving land production capacity and promoting the intensification and sustainable development of agricultural production.

**Keywords:** agricultural insurance; integrated disaster risk index; premium rate setting; regional differences; land production

## 1. Introduction

China is one of the countries with the largest number of natural disasters in the world. Droughts, floods, and typhoons occur frequently, causing serious impact on the national economy, especially agricultural production. As an important measure to safeguard agricultural production, agricultural insurance is highly valued. Carrying out risk assessment of agricultural production and determining premium rates induced by pure risk loss are important foundations for effectively guaranteeing the fairness, efficiency, and sustainability of agricultural insurance [1,2]. In 2015, the UN Summit formally adopted 17 Sustainable Development Goals, aiming to solve the development problems of social, economic, and environmental dimensions in an integrated way. The research on regional differentiated premium rates under the risk level of natural disasters has integrated the discussion of the above three dimensions, which has become an important task to promote the sustainable development of agriculture [3,4].

At present, many related studies have been carried out in the aspect of agricultural production risk assessment and are mainly evaluated through the synthesis of multiple indicators and the establishment of risk models [5–7]. One study constructed a mathematical model based on hazard, vulnerability analysis, and engineering defense capability to analyze the insurance risk degree of four natural disasters, including earthquake, geological disaster, flood, and typhoon [8], and other studies selected multiple indicators such as yield variability coefficient, drought and flood index, temperature anomalies, scale index,

and efficiency index to construct a risk model to measure and evaluate the production risk of vegetables and watermelon in Beijing [9]. Some studies chose drought risk degree, vulnerability index, yield risk index, and disaster resistance index from the perspective of meteorological factors, crop yield, and socioeconomic level to analyze and evaluate the drought risk of winter wheat in Henan Province [10]; some scholars evaluated the disaster risk of rice, wheat, and other major crops through a three-degree model of natural disaster risk [11,12]; and other scholars compared global major grid crop models to assess the risk of agricultural production in the context of climate change [13]. The former method can identify the relative level of drought risk with data being easy to obtain and calculate. However, the scientific nature of the index and weight restricts the accuracy of the evaluation. The latter based on the model has a high degree of quantification and the formation mechanism of risk is explored. However, the research on the loss caused by hazards remains to be deepened. This is also the development trend of future drought risk assessment.

In the aspect of the agricultural premium rate, the expected loss of crop yield is mainly calculated by the distribution pattern of regional crop yield so as to determine the theoretical pure rate [14–16]. Some studies fitted wheat trend yield by the Hodrick–Prescott filtering model and used the kernel density estimation method to initially determine wheat yield premium rates in various cities of Shandong Province [17]; other studies established premium rate models by coupling uniform rates with regional risk coefficients so as to calculate premium rates of corn [18]; some scholars using crop model pooling simulator, with county-level historical yield statistics, restored the loss rate of yield at the national 10 km grid level from 1991 to 2016 and finally determined the pure risk loss rate and catastrophe risk premium rate [19]; some studies have also proposed a general process for risk assessment and premium rate determination of agricultural insurance, which consists of collecting original data, estimating loss data, quantifying risk, spatially transforming risk quantification results, and calculating actuarial rates [16]; and many other scholars use methods of parameter, nonparametric kernel density, relative ratio, and loss cost ratio to assess crop yield risk and determine pure premium rates [20–24].

Generally, it can be seen that agricultural insurance and risk management are receiving more and more attention. Since the 1980s, a lot of research on agricultural risk assessment and the premium rate has been carried out, but it is not perfect, and the evaluation results are still uncertain. The main existing problems include data scarcity (such as insufficient data quantity or low data quality), fuzzy technology (such as instability caused by choosing different risk assessment models), the spatial mismatch between risk assessment and insurance pricing (such as inconsistency between the scale of risk assessment and premium rate), and insufficient specialization and refinement of the premium rate [16,25]. China is a country with a vast territory and the quality and quantity of data among regions vary greatly as well as the agricultural production situation and the ability to resist disaster [26]. Improving the credibility of the agricultural premium rates and insurance pricing in China has become the development direction of research.

However, the current agricultural insurance products cannot meet practical needs [27]. In this context, this paper adopts the detailed disaster loss data from 2015 to 2021 of flood, drought, and typhoon disasters obtained from the National Disaster Reduction Center of China (authoritative, complete, detailed, and long-term county-level disaster loss data to solve the problem of data quality to some extent), constructs a disaster index to express the integrated disaster risk levels, and established a quantitative model to determine the agricultural premium rate of each county (calculating the premium rate based on the losses, which enhanced the credibility of the result to a certain extent). This research aims to provide a reference for improving the accuracy of the agricultural premium rate and to create security for further improving land production capacity and promoting the intensification and sustainable development of agricultural production.

## 2. Materials and Methods

### 2.1. Basic Data

This study uses national disaster data from 2015–2021 to assess the integrated risk of agricultural natural disasters in various regions. Among the 11 major natural disasters affecting agriculture, floods, droughts, and typhoons cause greater losses, so the above three natural disasters are selected in this paper to assess the integrated disaster risk. The data include crop-affected areas, crop failure areas, and direct economic loss caused by the three disasters, and the data with high accuracy were from the only official disaster data management system called the National Disaster Management System (five-scale system of nation-province-city-county-township), which belongs to the National Disaster Reduction Center of the Ministry of Emergency Management of China (Table 1).

**Table 1.** Basic data.

| Name | Content | Source | Format | Size |
|---|---|---|---|---|
| Administrative divisions | China Administrative Divisions Map | National Geomatics Center of China | Vector | —— |
| Historical disaster data | Flood in China, 2015–2021 Drought in China, 2015–2021 Typhoon in China, 2015–2021 (Loss data of crop-affected area, crop-failure area, direct economic loss of each county) | National Disaster Reduction Center, Ministry of Emergency Management of China | Excel | 4793 records 1396 records 2894 records |

### 2.2. Research Methods

#### 2.2.1. Research Ideas and Framework

Based on the theoretical foundation of the regional disaster system, insurance, and geographic information mapping, as well as the database of disaster, geographic information, and policy literature, this study carries out research on agricultural risks and agricultural premium rates. The technical route is as follows (Figure 1).

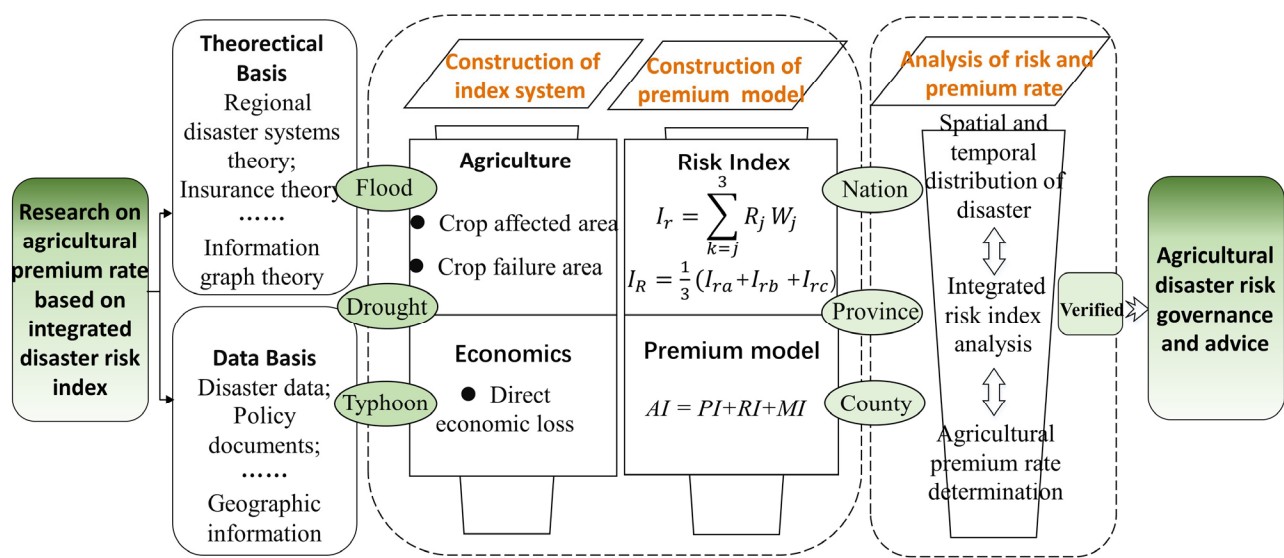

**Figure 1.** Research framework of agricultural premium rate based on disaster risk index.

Firstly, a disaster index system is established by selecting crop-affected areas, crop-failure areas, and direct economic losses caused by floods, droughts, and typhoons to describe the extent of agricultural losses; secondly, an integrated disaster risk index model

based on disaster data and a premium rate model are established. The former model is based on the accumulated historical data of multiple regions over many years, which are synthesized by a single-dimensional index to measure the average risk of agricultural damage and loss in a region, and the latter model is constructed on the basis of the risk index to combine with pure rate, risk surcharge rate, and administrative cost rate. Finally, the overall spatial and temporal distribution of disaster situation, the integrated disaster risk, and agricultural premium rate of provinces, cities, counties, and districts are analyzed. Based on the research results, the countermeasures and suggestions for agricultural disaster risk governance and insurance are further discussed.

2.2.2. Construction of Integrated Disaster Risk Index

Based on the literature review, it can be seen that the current agricultural production risk mainly includes assessment methods based on hazard factors, disaster loss, and risk mechanisms. The first type of method starts from the composition of disaster, which is simple to operate but highly subjective. The second kind of method starts from the risk result, the mathematical reasoning is strong but affected by the quality of data. The third type starts from the antecedents and consequences of risks, which can reveal the mechanism, but the modeling process is complicated [28]. Combined with reality, this paper refers to the second kind of method, considering the disaster loss, and constructs the integrated disaster risk index by the multi-index synthesis method and evaluates the average disaster loss degree of flood, typhoon, and drought in a region so as to represent the disaster risk level of the region. In this paper, we selected the indicators of crop-affected area, crop failure area, and direct economic loss to build the risk index. The weights of the indicators are determined by the methods of the entropy method and expert experience. The weight calculated by the entropy method is based on the information entropy of indicators: the greater the information entropy of an indicator, the greater the amount of information provided by the indicator and, therefore, the higher the weight should be. The method relies on the discrete degree of the data itself. The importance of the information of factors is known by the size of the entropy value, and the steps are as follows.

(1) Entropy method: Assuming that for *n* samples and *m* indicators, $x_{ij}$ is the observed value of the *jth* indicator of the *ith* sample and $x_{ij}$ is the normalized value:

   a.  Normalization of index:

$$x'_{ij} = \frac{x'_{ij} - min(x_j)}{max(x_j) - min(x_j)} \tag{1}$$

   b.  Calculation of information entropy redundancy:

$$P_{ij} = \frac{x_{ij}}{\sum_{i=1}^{n} x_{ij}} \tag{2}$$

$$E_j = \frac{\sum_{i=1}^{n} p'_{ij} ln(P_{ij})}{ln(n)} \tag{3}$$

$$D_j = 1 - E_j \tag{4}$$

$P_{ij}$ reflects the proportion of the *ith* sample value under the *jth* index, *i* = 1,..., *n*; *Ej* represents the entropy value of the index *j*, and *Dj* represents the redundancy degree of the index *j*, that is, the difference. *j* = 1,..., *m*;

   c.  Calculation of index weight:

$$W_j = \frac{D_j}{\sum_{i=1}^{n} D_j} \tag{5}$$

$W_j$ represents the contribution rate (weight value) of the index *j*, and the final weight value is obtained after appropriate revision by expert experience.

(2) Calculation of integrated disaster risk index:

$$I_r = \sum_{k=j}^{3} R_j W_j \tag{6}$$

$$I_R = \frac{1}{3}(I_{ra} + I_{rb} + I_{rc}) \tag{7}$$

$I_r$ represents the risk index of a disaster in the region, $R_j$ is the normalized value of the *jth* evaluation index, $W_j$ is the corresponding weight, $I_R$ is the integrated disaster risk index of the region, $I_{ra}$ is the flood risk index, $I_{rb}$ is the drought risk index, and $I_{rc}$ is the typhoon risk index.

It should be noted that, after sensitivity analysis and screening indicators, this paper uses the SPSS software tool to conduct principal component analysis on the initial indicators of crop-affected area, crop-failure area, grassland-affected area, forest-affected area, aquaculture-affected area, and direct economic loss, so as to obtain the variable load and determine the sensitivity coefficient of each indicator. Finally, combined with the integrity of the data, the index with sensitivity coefficients in the top three were selected to evaluate the agricultural disaster risk.

### 2.2.3. Establishment of Agricultural Premium Rate Model

It is generally believed that the premium rate (i.e., gross rate) consists of three parts: pure rate, risk surcharge rate, and administrative cost rate [11]. Among them, the pure rate is the risk loss rate, which is to make the premium income of the insurance company offset the expected compensation expenditure. In this paper, the integrated disaster risk index corresponds to the concept of the pure rate (the index is calculated based on the disaster data of crop-affected area, crop-failure area, and economic loss, which is used to characterize the degree of disaster loss). The risk surcharge rate mainly refers to the additional rate charged by insurance companies for controlling catastrophe overcompensation. The administrative cost rate refers to the rate charged by the insurer to the insured by including the administrative expenses which are incurred in carrying out the crop-insurance-related business. According to the above basic concepts, the models for calculating the agricultural premium rate are as follows:

$$AI = PI + RI + MI \tag{8}$$

$$PI = I_R * 80\% \tag{9}$$

$$RI = I_R * 2\% \tag{10}$$

$$MI = PI * 20\% \tag{11}$$

Among them, *AI* is the agricultural premium rate, *PI* is the pure rate, *RI* is the risk surcharge rate, *MI* is the administrative cost rate, and $I_R$ is the integrated disaster risk index.

In the actual calculation process, the pure rate is the pure risk loss rate with the same deductible terms (absolute and relative). The actual deductible conditions of agricultural insurance products varied in different regions. According to relevant research and experience, the pure rate will decrease rapidly with the increase in the relative deductible level. For every 10% increase in the relative deductible level, the corresponding pure rate may be reduced by 20% to 30%. Therefore, under this premise, if the relative deductible level is of 20% and above, the pure rate should be appropriately lower than the estimated premium rate of 80%. In order to enhance the rationality of the rate calculation in this paper, considering the average level of different deductible conditions of various products in the country, the pure rate is set to 80% of the integrated disaster risk index.

In this paper, the integrated disaster risk index is regarded as the expression of regional risk. On the basis of the literature, the maximum value of the risk surcharge is set at 2% of the integrated risk index [11,23]. (Note: in the classical insurance model, the risk surcharge rate is generally determined by the Probable Maximum Loss (*PML*) rule. In the process of determining *PML*, we must first know the exceedance probability curve of insurance compensation in a region, and this curve is obtained by adding the random variables that

represent the amount of compensation for each insurance contract according to the central limit theorem. Due to data limitations, here we use the integrated disaster risk index to replace the *PML* rule).

## 3. Results

### 3.1. Spatial and Temporal Distribution of Disaster

From 2015 to 2021, the losses of natural disasters (floods, droughts, and typhoons) suffered by the country over the years show a fluctuating downward trend (with slight ups and downs in between). The crop-affected areas ranked in the first three years were $4.8 \times 10^6$ hm$^2$ in 2015, $3.2 \times 10^6$ hm$^2$ in 2019, and $2.5 \times 10^6$ hm$^2$ in 2020, respectively. The crop-failure areas ranked in the first three years were $7.9 \times 10^5$ hm$^2$ in 2015, $4.9 \times 10^5$ hm$^2$ in 2016, and $4.6 \times 10^5$ hm$^2$ in 2019. The highest direct economic losses were 60.6 billion RMB in 2019, 58.2 billion RMB in 2017, and 57.8 billion RMB in 2015 (Figure 2).

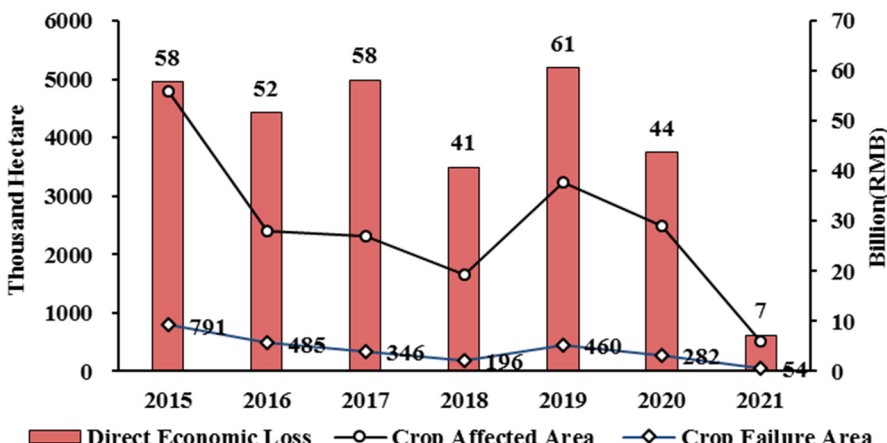

**Figure 2.** Disaster situation of natural disasters in China from 2015 to 2021.

From the view of the average damage caused by natural disasters in each county from 2015–2021, as a whole, most counties in Northeast and Central China and some counties in Southwest, North, and Northwest China were more severely affected. Specifically, 142 counties in North China were affected for more than 1000 hectares, with the top three counties being Horqinzuoyihou Banner, Jalaid Banner, and Horqinyouyihou Banner. A total of 110 counties in Northeast China had 1000 hectares or more affected and rounding out the top three counties are Yilan County, Shuangcheng District, and Dunhua City. There were 169 counties in East China with over 1000 hectares affected and ranking the top three counties are Lujiang County, Jin'an District, and Huoqiu County. A total of 114 counties in Central China had more than 1000 hectares of land affected and the top three counties are Luyi County, Tianmen City, and Jianli County. A total of 37 counties in southern China had an area of 1000 hectares or more affected and the top three counties are Lianjiang City, Leizhou City, and Meilan District. A total of 41 counties in southwest China had an area of 1000 hectares or more affected and the top three counties are Jinghong City, Mengla County, and Jiangcheng Hani and Yi Autonomous County. A total of 23 counties in northwest China had more than 1000 hectares affected and the top three counties are Tongwei County, Longxi County, and Huining County. The geographical regionalization of this paper (seven districts) is as follows: Northeast China includes Heilongjiang, Liaoning, and Jilin provinces; North China includes Beijing Municipality, Tianjin Municipality, Shanxi Province, Hebei Province, and Inner Mongolia Autonomous Region; East China includes Shanghai, Jiangsu, Zhejiang, Anhui, Jiangxi, Shandong, Fujian, and Taiwan provinces; Central China includes Henan, Hubei, and Hunan provinces; South China includes Guangdong Province, Guangxi Zhuang Autonomous Region, Hainan Province, Hong Kong Special Ad-

ministrative Region, and Macao Special Administrative Region; Southwest China includes Chongqing Municipality, Sichuan, Guizhou, and Yunnan provinces, and Tibet Autonomous Region. The northwest region includes Shaanxi, Gansu, and Qinghai provinces, Ningxia Hui Autonomous Region, and Xinjiang Uygur Autonomous Region (Figures 3 and 4).

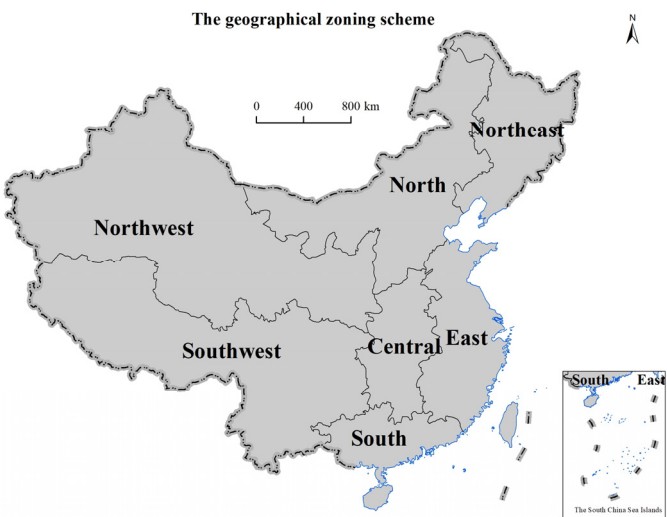

**Figure 3.** The geographic regionalization.

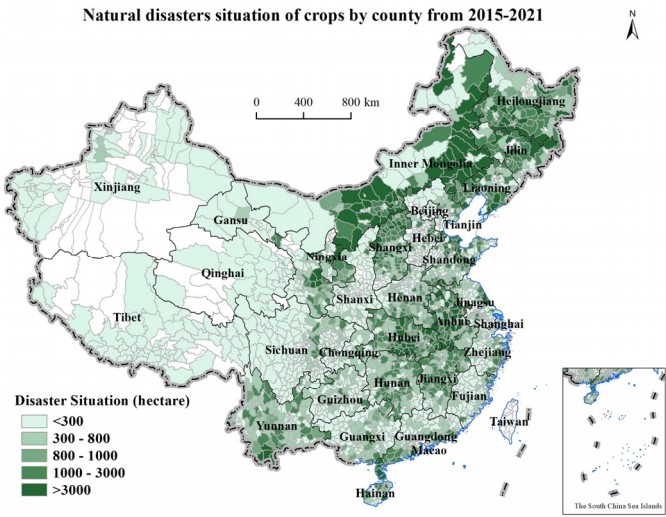

**Figure 4.** Natural disasters situation of crops by county from 2015–2021.

### 3.2. Analysis of the Integrated Disaster Risk

The national integrated disaster risk index is divided into five levels: extremely high risk (>0.1), high risk (0.03–0.1), medium risk (0.01–0.03), low risk (0.001–0.01), and extremely low risk (<0.001). The flood risk from 2015–2021 is generally characterized by a spatial distribution of high risk in central and northeast China, and some counties in Jilin Province, Inner Mongolia Autonomous Region, Hubei Province, Hunan Province, Anhui Province, and Jiangxi Province are in high flood risk areas. The drought risk generally shows a spatial distribution characterized by high risk in the north, central, and southwest, with some districts and counties in the provinces of Liaoning, Inner Mongolia, Shanxi, Hubei, Henan, Anhui, Jiangxi, Yunnan, Gansu, and Ningxia in high-drought risk. The typhoon risk generally shows the spatial distribution characteristics of a high index in coastal and northeastern regions, with coastal districts and counties in Shandong, Jiangsu, Zhejiang, Fujian, and Guangdong provinces, and some districts and counties in the junction

of Jiangsu, Anhui, and Henan as well as southern Heilongjiang in the high-risk area of typhoon hazard (Figure 5a–c).

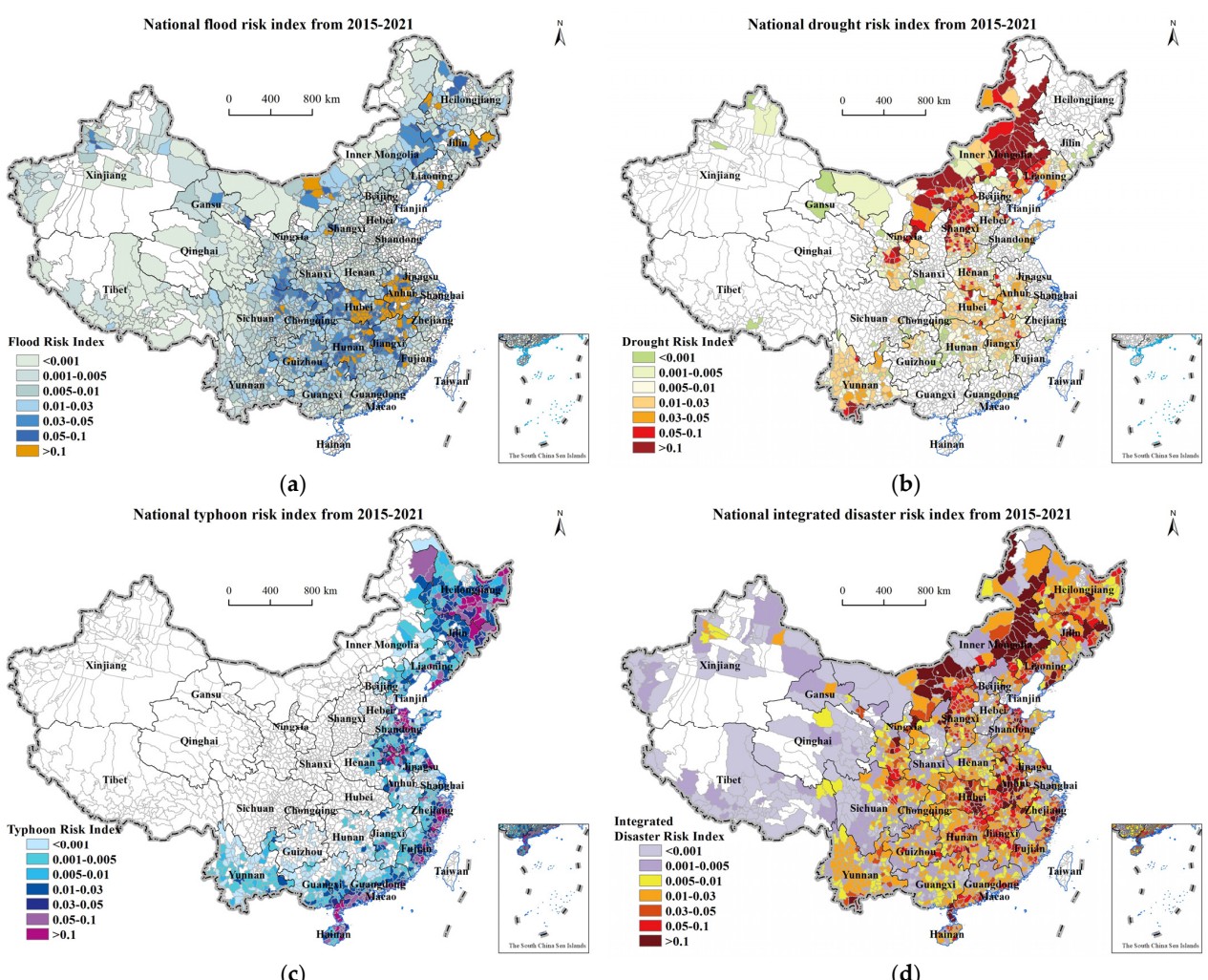

**Figure 5.** National flood (**a**), drought (**b**), typhoon (**c**), and integrated disaster (**d**) risk index.

The integrated disaster risk generally shows the spatial distribution characteristics of higher in the parts of northern, northeast, central, and eastern China. Some counties in provinces of Inner Mongolia, Shanxi, Jilin, Liaoning, Shandong, Zhejiang, Fujian, Anhui, Jiangxi, Hubei, Hunan, Guangdong, Yunnan, Guizhou, Sichuan, Gansu, and Ningxia are in high-risk areas of natural disaster. Nationally, there are 129 extremely high-risk counties, 392 high-risk counties, 633 medium-risk counties, 979 low-risk counties, and 593 extremely low-risk counties.

Specifically, there are 41 extremely high-risk counties in North China, and the top three are Horqinyouyihou Banner, Jalaid Banner, and Ar Horqin Banner with risk indices of 0.704, 0.657, and 0.469. There are 15 extremely high-risk counties in Northeast China, and the top three counties are Yongji County, Fengman District, and Dunhua City with risk indices of 0.38, 0.197, and 0.190. There are 43 extremely high-risk counties in East China, and the top three counties are Lujiang County, Shouguang City, and Linhai City, with risk indices of 0.374, 0.349, and 0.287. There are 14 extremely high-risk counties in Central China, and the top three counties are Ningxiang City, Tianmen City, and Qianjiang City, with risk indices of 0.253, 0.216, and 0.204, respectively. There are 9 extremely high-risk counties in South China, and the top three counties are Doumen District, Lianjiang City, and Xiangzhou District which have risk indices of 0.231, 0.199, and 0.185. There are 2 extremely high-risk

counties in southwest China, namely, Mengla County and Jinghong City, and their risk indices are 0.136 and 0.115. There are 5 extremely high-risk counties in northwest China, and the top three are Yanchi County, Zizhou County, and Suide County, and their risk indices are 0.175, 0.146, and 0.106, respectively (Figure 5d).

### 3.3. Determination of Agricultural Premium Rate

National agricultural premium rates based on the integrated disaster risk index are divided into 5 grades of extremely high rate (>0.15), high rate (0.08–0.15), medium rate (0.06–0.08), low rate (0.02–0.06), and extremely low rate (<0.02). From 2015 to 2021, the overall distribution characteristics of the agricultural premium rates of counties affected by flood were similar to that of the flood risk index. In addition, some counties in eastern Jilin, central Inner Mongolia, eastern and southern Hubei, southern and northeastern Hunan, most of Anhui, and central and northeastern Jiangxi Province are in the high-level of the flood agricultural premium rates. The overall distribution pattern of agricultural premium rates of drought is similar to that of the drought risk index. Some counties in northeastern and central Inner Mongolia, northern Shanxi, southeastern Gansu, eastern Ningxia, and southern Yunnan are at a high level of drought agricultural premium rates. The overall distribution law of agricultural premium rates of typhoon is similar to that of the typhoon risk index, with high agricultural premium rates in some counties in Shandong, Zhejiang, Fujian, Guangdong, Heilongjiang, Jiangsu, Anhui, and Henan (Figure 6a–c).

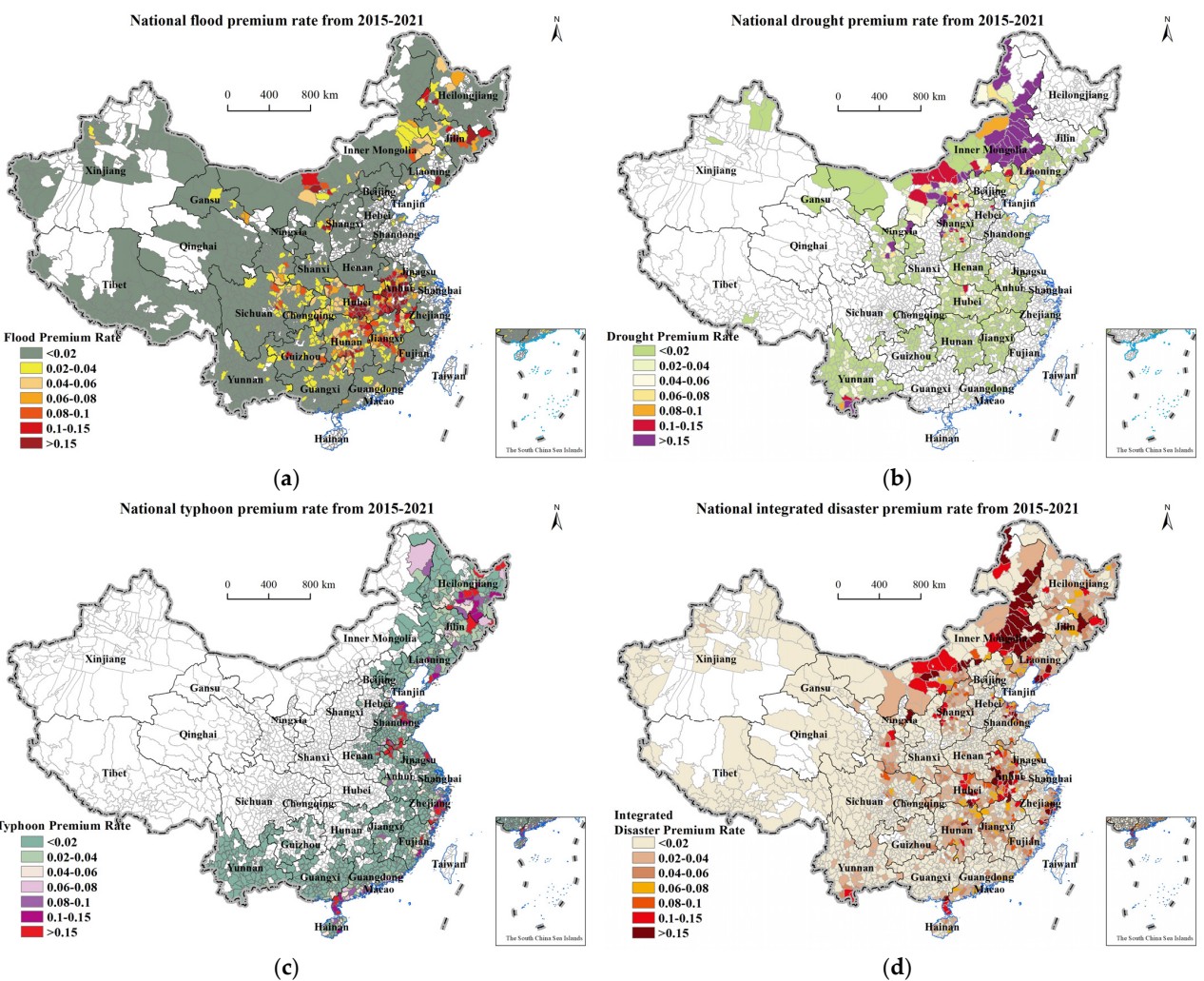

**Figure 6.** National flood (**a**), drought (**b**), typhoon (**c**), and integrated disaster (**d**) premium rates.

The overall distribution pattern of agricultural premium rates of natural disasters by county from 2015–2021 is similar to that of the integrated disaster risk index, with some counties in Inner Mongolia, Shanxi, Liaoning, Jilin, Shandong, Anhui, Jiangxi, Zhejiang, Fujian, Guangdong, Hubei, and Hunan provinces having high agricultural premium rates. Of the 5 rate grades, there are 63 extremely high-rate counties, 97 high-rate counties, 71 medium-rate counties, 498 low-rate counties, and 1997 extremely low-rate counties. (For a more flexible reference effect, we provide rate intervals for each county based on the only results calculated; for more information, please contact the research team).

Specifically, there are 24 extremely high-rate counties in North China, and the top three are Horqinyouyiqian Banner, Jalaid Banner, and Ar Horqin Banner, with rates of 0.670, 0.644, and 0.460. There are 6 extremely high-rate counties in Northeast China, and the top three are Yongji County, Fengman District, and Dunhua City, with rates of 0.371, 0.193, and 0.186. There are 23 extremely high-rate counties in East China, and the top three are Lujiang County, Shouguang City, and Linhai City, with rates of 0.366, 0.342, and 0.281. In Central China, there are 6 extremely high-rate counties, and the top three are Ningxiang City, Tianmen City, and Qianjiang City, with rates of 0.248, 0.212, and 0.200. In South China, there are 3 extremely high-rate counties, namely, Doumen District, Lianjiang City, and Xiangzhou District, with rates of 0.226, 0.195, and 0.181. There are no extremely high-rate counties in Southwest China. There is only 1 extremely high-rate county in Northwest China, Yanchi County, with a rate of 0.171 (Figure 6d).

## 4. Discussion

This paper calculates agricultural premium rates of counties based on disaster risk levels, which provides a reference for high-precision agricultural premium rate setting. In the future, further work can be carried out in the following areas:

(1) Enhance the credibility of premium rates

Faced with the existing problems in agricultural risk assessment and premium rate research, such as data scarcity, space mismatch between risk assessment and insurance pricing, and insufficient specialization and refinement of premium rates, many scholars have carried out a large number of studies to solve the above problems. For example, some studies have proposed to add meteorological information [29] and soil information [30] to make up for the lack of data, and some studies have also provided the regional scale displacement model [23] and the econometric model of data deviation to solve problems such as the spatial scale mismatch between risk assessment and agricultural insurance pricing [31].

On the basis of the previous studies, this paper collects authoritative, complete, detailed, and long-term county-level disaster loss data and applies more than 9000 records to calculate the rate, which solves the problem of quantity and quality of data to a certain extent. The constructed disaster index expressed the integrated disaster risk of flood, drought, and typhoon, fully considered the loss of disaster-bearing body, and improved the accuracy of agricultural risk assessment to a certain extent. Based on the premium rate result and the actual situation of agricultural insurance, the reference range of premium rate in each county is given, which improves the scientificity of premium rates. In the near future, big data technology should be deeply explored and a variety of data resources should be integrated to further enhance the credibility of premium rate determination.

(2) Optimize agricultural insurance products

Index insurance is an innovative agricultural insurance product that differs from traditional insurance based on damage payouts. It is not based on the actual loss suffered by the specific subject matter of the insurance, but rather on the index agreed upon in the insurance contract, which is getting more and more attention. For example, some studies chose accumulated precipitation, accumulated temperature, precipitation intensity, and frequency to design weather index insurance products [32,33], and some studies used Grach, Egarch, Copula, and other models to simulate the distribution of the weather index [34,35] and the process of considering the maximum stability of extreme weather

variables in index products [36]. Therefore, index-based products develop rapidly and have a great advantage in developing countries, especially for those with a large number of farmers and decentralized agricultural production operations.

However, agricultural index insurance is heavily influenced by a single variable and is characterized by a high dependence on indicators. The most widely used regional yield insurance and weather index insurance currently face challenges in risk pricing including insufficient empirical data, complicated dependencies between different risks, and so on [37]. In future research, it is suggested to further integrate multi-source and multi-scale Earth observation data (including satellites, drones, Earth surface information, and network data) [4], establish an intelligent index model, and simulate and analyze farmers' economic losses and insurance compensation scenarios so as to improve the scientific nature of the claim mechanism and promote sustainable development of agricultural insurance.

(3)    Promote the study of insurance regionalization

Insurance regionalization is important for promoting the specialization and refinement of agricultural insurance. Some scholars put forward the issue of risk regionalization and premium rate regionalization of crop insurance in 1994, leading to a series of explorations of the research work concerned with agricultural insurance regionalization [38]. Some scholars built a quantitative model of agricultural insurance risk and pricing, which laid the foundation for insurance regionalization [39]. The former China Insurance Regulatory Commission has set up research projects on plantation insurance regionalization at the national–provincial and provincial–county levels [40,41]. The central government of China has issued several documents to promote the high-quality development of agricultural insurance, making it an urgent need to speed up this fundamental work, which can be explored further as follows:

In terms of research methodology, a set of suitable technical methodologies will be developed by applying regional natural disaster risk assessment, premium rate determination, regional differentiation theory, and quantitative methods, relying on mathematical statistics, simulation, machine learning, and other technical means. Regarding spatial accuracy, agricultural production risk assessment and premium rate regionalization can be implemented at the county level, with further refinement at the village level in advance. For crop categories, the focus will be on the three major food crops (rice, wheat, and maize), for which the new rate scheme is being implemented on a trial basis, and we will gradually carry out the risk assessment, rate setting, and regional division for insurance products with high loss rates and high risks, such as bulk food and oil crops [23]. Based on the above work, the preparation of agricultural production risk regionalization maps will be expedited to provide stronger scientific and technical support for agricultural insurance so as to create security for further improving land production capacity [42,43].

(4)    Improve the risk-sharing mechanism among farmers, insurance enterprises, and the government

Agricultural production is an important source of income for hundreds of millions of farmers and a basic guarantee for national food security and sustainable development. Agricultural disaster risk governance must rely on the joint participation of farmers, the government, and insurance enterprises ("farmers-government-insurance enterprises" governance system) to realize risk and benefit sharing. The government shall increase investment in disaster reduction planning and safety construction, strengthen support to insurance enterprises for the planting industry, and provide premium subsidies and post-disaster relief to farmers. Insurance enterprises should strengthen the risk management of the planting industry and form market power through "joint venture" and "transfer" businesses. Farmers should further strengthen their risk management level at the field scale, and form "safe communities" to effectively reduce crop and income loss caused by disasters. Base on the paradigm of risk sharing, the agricultural insurance can achieve a state of better service, more efficient operation, and more scientific management so as to improve the resilience of land production and ensure national food security.

## 5. Conclusions

Generally, since the period of 2015–2021, the national natural disaster situation has remained complicated and severe. Extreme weather and climate events occur frequently, and floods, droughts, and typhoons are still the main disasters affecting agriculture. Based on county-level disaster data, this study analyzed the regional differences in disaster loss, disaster risk, and agricultural premium rates across the country by establishing an integrated disaster risk index and an agricultural premium rate model, further improving the accuracy of agricultural premium rates and providing a reference for the implementation of region-differentiated rates across the country:

(1)  From the perspective of disaster loss, generally speaking, it has the distribution characteristics of spatial agglomeration, which shows that most counties in Northeast and Central China, some counties in Southwest, North, and Northwest China experienced relatively serious loss. There are 636 counties in the country with an area affected of more than 1000 hectares. Among them, Horqinzuoyihou Banner in North China, Yilan county in Northeast China, and Luyi County in Central China were severely affected.

(2)  From the perspective of integrated disaster risk level, the risk in most parts of Northern, Northeast, Central, and Eastern China is relatively high. Among the 5 risk grades, there are 129 extremely high-risk counties, 392 high-risk counties, 633 medium-risk counties, 979 low-risk counties, and 593 extremely low-risk counties. Horqin Youyiqian Banner, Jalaid Banner, and Ar Horqin Banner in North China, Yongji County in Northeast China, and Lujiang County in East China have extremely high integrated disaster risk, with indices being 0.704, 0.657, 0.469, 0.378, and 0.374, respectively.

(3)  From the perspective of the agricultural premium rate, it is similar to the distribution pattern of the integrated disaster risk. Agricultural premium rates are at a high level in some counties in Inner Mongolia, Shanxi, Liaoning, Jilin, Shandong, Anhui, Jiangxi, Zhejiang, Fujian, Guangdong, Hubei, and Hunan. Among the 5 rate-grade areas, there are 63 extremely high-rate counties, 97 high-rate counties, 71 medium-rate counties, 498 low-rate areas, and 1997 extremely low-rate counties. This is closely related to the risk level of the Horqin Youyiqian Banner, Jalaid Banner, and Ar Horqin Banner in North China, Yongji County in Northeast China, and Lujiang County in East China, which have extremely high agricultural premium rates, with indices of 0.670, 0.644, 0.460, 0.371, and 0.366. The premium rates in this study can be divided into reference ranges for rates based on practical situations.

Due to the limited conditions, there are still some deficiencies in this study. For example, when calculating the integrated risk of natural disasters, cold wave, heat wave, and snow disaster are not considered. Furthermore, assessment units can be further refined, extending from county level to grid scale. The risk of crop production (such as wheat, rice, and maize) should be further considered on the basis of disaster risk, and thus linked to current insurance products.

**Author Contributions:** Conceptualization, Y.W. and G.G.; methodology, L.F. and Y.W.; software, Y.W.; validation, H.L., L.F. and G.G.; formal analysis, Y.W.; data curation, Y.W.; writing—original draft preparation, Y.W. and H.L.; writing—review and editing, H.L., L.F. and G.G.; visualization, Y.W.; supervision, G.G.; project administration, G.G. All authors have read and agreed to the published version of the manuscript.

**Funding:** This research received no external funding.

**Acknowledgments:** This work was supported by the Science and Technology Project of Shanghai Tai'an Agricultural Insurance Institute.

**Conflicts of Interest:** The authors declare no conflict of interest.

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
