# Peer review of "Quantitative Study on Agricultural Premium Rate and Its Distribution in China"

_land, doi:10.3390/land12010263_

Round 1

Reviewer 1 Report

I think this paper provides very interesting aspect and valuable information on the relationship between assessment of integrated risk and agriculture insurance premium rate. The novelty of this study lies on the country level data analyses, and the emphasis of relying on integrated risk assessment tool to determin insurance preimum rate. Being that said, I believe the manuscript needs the following revisions to provide logically clear and scientifically sound argument.

First, the literature review simply present lits of the papers that focus on risk governance and models, though the authors needs to summarize the advantages and disadvantages of these methods, especially the shortcomings of the methods to why it is important to develop more integrated risk assessment framewor, which I believe your manuscript tends to highlight.

Second, the justification for importance of the study needs further improvement.

Third, discussion section of the paper focus more on recommendation for future studies rather than discussing the findings of the study in the manuscript. The authors need to discuss the importance and contribution of the findings in their studies, and how do these findings contribute to the existing studies.

Finally, the English language requires major edits and language polishing. Many of the sentences are too long to decipher.

Author Response

Dear review, we have revised the article according to your  suggestiongs. Please refer to the notes and revised version of this article for details. Thank you! 

Reviewer 2 Report

This study develops a quantitative model to determine the agricultural premium rate for each county in China using disaster data from three major natural disasters, flood, drought and typhoon, from 2015 to 2021. The paper is of a high standard and acceptable after minor revision. 

Minor Comments:

1. The study is of high quality. Thus, I recommend securing English editing services to address the grammar issues and improve the language and presentation.

2. Line 11: Please use a sentence to clearly present the primary aims of the study. The sentence could be: This study develops a quantitative model to .... 

3. Lines 33 and 319: Please add a sentence to explain the relationship between the issue of the study and sustainable development, such as the Sustainable Development Goals (SDGs). Especially, it is better to highlight the contribution of Earth observations to sustainable development by referring https://doi.org/10.3390/rs13081528. 

4. Section 2.2: It is better to add a subsection of the sensitivity analysis of the developed methods. Model validation is essential for any study. From my perspective, sensitivity analysis is a reasonable approach to model validation for the developed indexes in this study. It is not necessary to conduct a complex sensitivity analysis because it is not the focus of the study. A brief sensitivity analysis is required. 

Author Response

Dear reviewer, we have revised the article according to your  suggestions. Please refer to the notes and the revised version of this article for details. Thank you! 

Reviewer 3 Report

This manuscript is well written, easy to read and the topic is very useful for agriculture insurance. I recommend to publish after minor changes. I noticed places to improve:

1. line 36: at home and abroad --> not appropriate need better words or delete

2. line 49: still -->also

3. line 72: realistic needs --> practical need ?

4. Figure1 --> Figure 1. All figures have the same problem.

5. Line 185 : Crop failure are --> crop failure area

6. Line 205: The high risk (>0.1) has larger number than higher risk(0.03-0.1). This is not right. The five levels should be like:   Low, Medium, High, and Extremely High

7. The figure legends are not clear for figures with map.

Search Results

Featured snippet from the web

Search Results

Featured snippet from the weLow, Medium, High, and Extremely High

Author Response

(The authors gave the same response as above.)

Reviewer 4 Report

The paper is interesting and the conducted study has significant importance for the case of China agriculture.

In the Introduction section [lines 52- 53]  it is written „In the aspect of agricultural premium rate, the expected loss of crop yield is mainly calculated by the distribution pattern of regional crop yield and the corresponding guarantee level of yield”. I do assume that the guarantee level of yield is characteristic for the agricultural policy of ChRP. In free market driven policies there are no guarantees for the level of yield. It is recommended thus to explain the specific situation of the country under discussion. And relate that to the situation of other cointries, with regard to the usefullnes of the method proposed.

The methodological approach proposed requires relation to the existing body of knowledge. There is no enough literature analysis in the paper, esp. in the part [116-122]. There should be provided references that will show if proposed approaches have been already used, what are they limitations. It is not clear if the approach described in lines 142-153 has been already used in other studies or not. To write :it is generally believed that the premium rate (i.e. gross rate) consists of three parts: pure rate, risk surcharge rate and administrative cost rate." [142-143] you need to provide references. The literature review is need.

The discusson section is made based on the Autors' own opinions, but not faced with the results of other studies. Thus requires re-writing .

In the conclusion section there ar any limiattions described, thus such should be added [364].

I suggest minor review.

Author Response

(The authors gave the same response as above.)
